# The Role of *SNAP-25* in Autism Spectrum Disorders Onset Patterns

**DOI:** 10.3390/ijms241814042

**Published:** 2023-09-13

**Authors:** Elisabetta Bolognesi, Franca Rosa Guerini, Alessandra Carta, Matteo Chiappedi, Stefano Sotgiu, Martina Maria Mensi, Cristina Agliardi, Milena Zanzottera, Mario Clerici

**Affiliations:** 1Laboratory of Molecular Medicine and Biotechnology, IRCCS Fondazione Don Carlo Gnocchi, Via Capecelatro 66, 20148 Milan, Italy; ebolognesi@dongnocchi.it (E.B.); cagliardi@dongnocchi.it (C.A.); mzanzottera@dongnocchi.it (M.Z.); mario.clerici@unimi.it (M.C.); 2Unit of Child Neuropsychiatry, Department of Medicine, Surgery and Pharmacy, University of Sassari, 07100 Sassari, Italy; carta.ale84@gmail.com (A.C.); stefanos@uniss.it (S.S.); 3Child Neuropsychiatry Unit, IRCCS Mondino Foundation, 27100 Pavia, Italy; mchiappedi@gmail.com (M.C.); martina.mensi@mondino.it (M.M.M.); 4Department of Pathophysiology and Transplantation, University of Milan, 20122 Milan, Italy

**Keywords:** autism spectrum disorders (ASD), ASD classic onset, ASD regressive onset, *SNAP-25* gene (synaptosomal-associated protein of 25 kDa), regression, epigenetics

## Abstract

Autism spectrum disorders (ASD) can present with different onset and timing of symptom development; children may manifest symptoms early in their first year of life, i.e., early onset (EO-ASD), or may lose already achieved skills during their second year of life, thus showing a regressive-type onset (RO-ASD). It is still controversial whether regression represents a neurobiological subtype of ASD, resulting from distinct genetic and environmental causes. We focused this study on the 25 kD synaptosomal-associated protein (*SNAP-25*) gene involved in both post-synaptic formation and adhesion and considered a key player in the pathogenesis of ASD. To this end, four single nucleotide polymorphisms (SNPs) of the *SNAP-25* gene, rs363050, rs363039, rs363043, and rs1051312, already known to be involved in neurodevelopmental and psychiatric disorders, were analyzed in a cohort of 69 children with EO-ASD and 58 children with RO-ASD. Both the rs363039 G allele and GG genotype were significantly more frequently carried by patients with EO-ASD than those with RO-ASD and healthy controls (HC). On the contrary, the rs1051312 T allele and TT genotype were more frequent in individuals with RO-ASD than those with EO-ASD and HC. Thus, two different *SNAP-25* alleles/genotypes seem to discriminate between EO-ASD and RO-ASD. Notably, rs1051312 is located in the 3′ untranslated region (UTR) of the gene and is the target of microRNA (miRNA) regulation, suggesting a possible epigenetic role in the onset of regressive autism. These SNPs, by discriminating two different onset patterns, may represent diagnostic biomarkers of ASD and may provide insight into the different biological mechanisms towards the development of better tailored therapeutic and rehabilitative approaches.

## 1. Introduction

Autism spectrum disorders (ASD) are part of the neurodevelopmental disorders (NDDs) classified by the Diagnostic and Statistical Manual of Mental Disorders-5, (DSM-5) [1] as a group of conditions characterized by dysfunctional deviation of the neurodevelopmental trajectory, causing impairments of cognitive and behavioral functioning. ASD are among the commonest of NDDs, affecting 1–2% of the population, and are approximately four times more common in males than females [2,3]. ASD are characterized by a clinical spectrum of difficulties in social communication and interaction with restrictive and repetitive patterns of behaviors and interests. ASD are genetically heterogeneous with a multitude of implicated genes [4,5,6,7].

The clinical signs of ASD may have a different timing of appearance. Some children may manifest symptoms as early as within their first year of life and are considered as patients with early onset (EO-ASD). Others may lose already achieved skills, i.e., communication, during their second year of life, thus showing a regressive-type onset (RO-ASD). This developmental regression is estimated in 32% of children with ASD [8], involving decays in the main social and communication behaviors for most of them [9].

It is still not clearly established whether regressive ASD may be a neurobiological subtype with genetic and environmental susceptibility factors distinguishable from those of EO-ASD [10]. The biological mechanisms that lead to the regressive phenotype are also still unknown.

Regression has been associated with the occurrence of seizures [11], brain growth [12,13], immunological function [14,15], gastrointestinal problems [16], and genetic and genomic variations [17,18,19] that include mitochondrial and MeCP2 mutations [20,21,22]. A genetic involvement in the ASD-regressive phenotype has been suggested specifically for those genes that play a role in synaptic transmissions, such as SHANK3 and SYNGAP1 [23]. Goin-Kochel et al. [17] reported that children with ASD and mutations in postsynaptic density genes were more likely to experience regression than those with mutations in embryonic genes who were less likely to have skill losses.

Among the genes that play a role in synaptic plasticity, the *SNAP-25* (synaptosomal-associated protein of 25 kDa) gene has been largely associated with common neuropsychiatric disorders, such as attention deficit/hyperactivity disorder (ADHD) [24,25], bipolar disorders, and schizophrenia [26].

The *SNAP-25* rs363043 and rs363050 polymorphisms were shown to be associated with hyperactivity and cognitive deficits in Italian children with ASD, respectively [27,28], whereas in the Han Chinese population, the rs363018 and rs8636 polymorphisms predicted a higher likelihood of developing ASD [29]. Finally, the rs3746544 G allele was observed to be more frequently carried by Iranian children with ASD [30]. In addition, the *SNAP-25* gene has been largely associated with different neurological and non-neurological disorders. *SNAP-25* rs363050 (A) and rs363043 (T) alleles as well as the rs363050/rs363043 A-T haplotype were associated with Alzheimer’s disease [31]. Similarly, the rs363050 AA genotype was significantly associated with the risk of sarcopenia, [32] whereas *SNAP-25* rs1051312, was suggested to play a role in the development of multiple sclerosis [33].

*SNAP-25*, *VAMP* (vesicle-associated membrane protein), and syntaxins constitute the SNARE complex; these proteins are essential for the exocytosis of synaptic vesicles and, thereby, for synaptic transmission [34].

The presynaptic role of *SNAP-25* in regulated exocytosis is well established; however, recently, a postsynaptic role of *SNAP-25* has also been reported, including involvement in spine morphogenesis [35,36] and structural changes necessary for long-term potentiation (LTP) and synaptic maturation [37,38]. Recently, its localization in the postsynaptic plasma membrane of the hippocampus was demonstrated [39].

The principal aim of this study consisted of an evaluation of the possible role of the *SNAP-25* gene in differentiating children with EO-ASD from those who develop a regressive form of ASD. To this purpose, four single nucleotide polymorphisms (SNPs) of the *SNAP-25* gene (i.e., rs363043, rs363039, rs363050, and rs1051312), known players in neurodevelopmental disorders, were investigated [24,25,26,27,28,29,30].

## 2. Results

### 2.1. SNAP-25 rs363050, rs363039, rs363043, and rs1051312 Genotype Distribution in Children with ASD and Healthy Controls

The allele and genotype distributions of *SNAP-25* rs363050, rs363039, rs363043, and rs1051312 in children with ASD and HC are shown in Table 1. Genotype frequencies for each polymorphism were in Hardy–Weinberg equilibrium both in ASD and in controls. First, the *SNAP-25* rs363050, rs363039, rs363043, and rs1051312 allelic and genotypic distributions in children with ASD from continental Italy and in those from Sardinia were compared, and since no differences were observed, the two cohorts were grouped. The frequency of the *SNAP-25* SNPs minor alleles observed in the control group was in agreement with previously reported results [27]. A higher frequency of the rs363039 G major allele was found in children with ASD compared to controls (70% vs. 63%, *p* = 0.05, OR = 1.4, 95% IC (0.99–1.95)), though not statistically significant. No further difference in allelic and genotype frequencies was observed for the other polymorphisms.

### 2.2. SNAP-25 rs363050, rs363039, rs363043, and rs1051312 Allele and Genotype Distributions in Children with ASD with Early and Regressive Onsets

Comparisons of the allele and genotype distributions were investigated in the EO-ASD, RO-ASD, and HC groups. The rs363039 G allele was more frequently carried by children with EO-ASD compared to both the children with RO-ASD (75% vs. 65%, *p* = 0.08, not significant) and HC (75% vs. 63%, *p* = 0.009, OR: 1.75, 95% IC (1.14–2.72)). Notably, the allelic frequency in the children with RO-ASD is not dissimilar to that of the HC group.

On the other hand, the rs1051312 T allele was more frequently carried by children with RO-ASD than those with EO-ASD (87% vs. 72%, *p* = 0.003, OR: 2.64, 95% IC (1.38–5.22)) and the HC group (87% vs. 73%, *p* = 0.002, OR: 2.43, 95% IC (1.38–4.5)). The genotype and allelic distributions of rs1051312 were similar between the children with EO-ASD and the HC.

The co-dominant model distribution among the studied groups confirmed the results observed for the allelic ones, as reported in Table 2. The rs303039 GG genotype is more frequent in the EO-ASD group compared to the RO-ASD (58% vs. 40%) and HC groups (58% vs. 38%). Again, only the latest comparison is statistically significant (*p* = 0.015). Conversely, the rs1051312 TT genotype is significantly more frequent in the RO-ASD group compared to the EO-ASD (76% vs. 49%, *p* = 0.008) and HC (76% vs. 53%, *p* = 0.006) groups.

In Table 3, the results are reported when we considered different genetic models. We found that, under the Dominant model, the frequency of the rs363039 GG genotype was higher in the EO-ASD group than in the RO-ASD group (58% vs. 40%, *p* = 0.04, OR: 2.19 (1.03–4.27)), and the rs1051312 TT distribution was higher in the children with RO-ASD than that in the children with EO-ASD (76.0% vs. 49.0%, *p* = 0.002, OR: 3.2, (1.5–6.9)).

### 2.3. Haplotype Analysis of SNAP-25 rs363050, rs363039, rs363043, and rs1051312 in the ASD Early and Regressive Onsets and Healthy Controls Groups

A haplotype analysis was conducted to evaluate if different *SNAP-25* haplotypes are associated with the different ASD onsets. As reported in Table 4, the *SNAP-25* rs363050, rs363039, rs363043, and rs1051312 A-G-T-C haplotype out of the 11 haplotypes that are generated from their combinations confers a 4.7-fold risk to have EO-ASD rather than RO-ASD (17.5% vs. 4.2%, *p* = 0.0008). Interestingly, the RO-ASD group has a frequency of this haplotype similar to the HC (4.2% vs. 7.2%). On the contrary, the G-A-C-T haplotype is more frequent in the RO-ASD group than in the EO-ASD group (31.0% vs. 20.3%), but the difference is not statistically significant. When we restricted the analysis to the haplotypes of rs363039 and rs1051312, the G-C haplotype was a risk factor for EO-ASD (*p* = 0.004, OR: 2.7 (1.3–5.5)), while the A-T haplotype was a risk factor for RO-ASD (*p* = 0.02, OR: 1.8 (1.1–3.3)).

Finally, Figure 1 shows the results of the linkage disequilibrium (LD) analysis for the *SNAP-25* SNPs. The linkage analysis showed a strong linkage disequilibrium (LD) among rs363050, rs363039, and rs363043 (D’ > 90), but no linkage disequilibrium between them and rs1051312 (D’ < 50). Rs1051312 and rs363039 can be considered independent markers in the *SNAP-25* gene.

### 2.4. Correlation of Clinical and Functioning Scales with the ASD Early Onset, Regressive Onset, and SNAP-25 Genotypes

We analyzed the possible correlations among clinical scales, different onset patterns, and *SNAP-25* polymorphisms. First, we compared ASD severity and cognitive and functioning scores between the EO-ASD and RO-ASD groups. The only significant result was a worse cognitive level in the EO-ASD group than in the RO-ASD group (ANOVA: mean values 2.4 ± 1.2 vs. 1.9 ± 1.3, *p* = 0.038). The cognitive level was grouped according to the historically used severity scale for mental retardation (as in DSM IV-TR, [40]) with scores from 1 to 5 (1 = normal (IQ >85), 2 = borderline IQ (70 to 85), 3 = mild mental retardation (IQ 50–55 to 70), 4 = moderate retardation (IQ 35–40 to 50–55), 5 = severe mental retardation (IQ: 20–25 to 35–40), profound mental retardation (IQ: <20–25)). When analyzing this association by adding the *SNAP-25* rs363090 and rs1051312 genotypes, we found no differences in the cognitive levels between the EO-ASD and RO-ASD groups. These results suggest that the difference in the cognitive scores of the EO-ASD and RO-ASD groups is independent of the *SNAP-25* polymorphisms.

## 3. Discussion

Our study tried to shed some light on the different onset patterns of ASD by comparing genetic variants of *SNAP-25* in children with early- and regressive-onset ASD. Children with the EO-ASD pattern have evident neurodevelopmental delays and deviances already in their first year of life, while children with the regressive pattern have an apparent typical neurodevelopment until they initiate a decline or loss of the acquired skills. For many authors, regression in ASD represents a neurobiological subtype with different causes from the early-onset type [10,41].

*SNAP-25* is a protein involved in pre- and post-synaptic functions. Our results showed that the *SNAP-25* rs363039 G allele and GG genotype were significantly more frequent in the EO-ASD group than in the RO-ASD group in which, remarkably, the frequencies are similar to those of the HC.

On the contrary, in the RO-ASD group, the rs1051312 T allele and TT genotype have a higher frequency than in the early onset group, which has, in turn, similar frequencies to the HC.

The haplotype analysis showed that the rs363050, rs363039, rs363043, and rs1051312 haplotype A-G-T-C confers a 4.7-fold higher risk to develop EO-ASD. More specifically, the rs363039, rs1051312 G-C haplotype is significantly associated with EO-ASD, whereas the rs363039, rs1051312 A-T haplotype seems to characterize RO-ASD.

Therefore, rs363039 and rs1051312 can discriminate between EO-ASD and RO-ASD. Noteworthy, the linkage disequilibrium analysis demonstrated that they are independent markers that are not linked to each other. It is important to underline that rs1051312, located in the 3′UTR of the gene, represents a putative target site for the miR-641 [42], and the T allele is its target. MicroRNAs (MiRNA) are short, noncoding mRNA that are involved in epigenetic regulation and affect protein levels at the post-transcriptional level [43]. We may hypothesize that, in ASD regression, an adverse environmental factor can trigger an epigenetic dysregulation that alters *SNAP-25* expression, resulting in disease-associated misregulation of this protein and causing the decline or loss of the acquired neurodevelopmental skills. Several other miRNAs were suggested to modify *SNAP-25* expression, such as miR-153 and miR-27 (down) [44,45], miR-23a-3p, and miR-181a-5p [46].

A few studies tried to differentiate individuals with regressive and non-regressive ASD, associating distinct neural phenotypes with different onsets. Nordhal et al. [12], by using MRI techniques, analyzed the total cerebral volume in relation to ASD onset and found that boys with regression-type ASD had enlarged brain volumes compared to both boys without regression and typically developing matched controls. They also found that head growth trajectory was significantly accelerated in male children with RO-ASD (not in females) compared to both boys with EO-ASD and HC, beginning at approximately the 4/6 month of age and proceeding to the onset of behavioral symptoms [47].

Enstrom et al. [48] found different plasma levels of the brain-derived neurotrophic factor (*BDNF*), a critical factor for neuronal differentiation and synaptic development, in children with ASD with different clinical onset. BDNF levels were significantly higher compared with age-matched controls without autism. Interestingly, within the ASD group, children with EO-ASD had higher plasma levels of BDNF compared to children with RO-ASD who, in turn, showed comparable BDNF levels with normal developing children, suggesting, within the ASD group, a differential BDNF response based on the onset of the autistic behaviors.

Li X et al. [49] found higher plasma levels of total secreted amyloid precursor protein (sAPPtotal) and of secreted amyloid precursor protein-α (sAPPα) in children with RO-ASD compared to children with non-regressive ASD and typically developing controls. APP is a glycoprotein secreted by glial cells and neurons that promotes neuronal proliferation and migration, cell adhesion, and synapse formation. At a young age, high levels of APP can reduce developmental spine pruning and impair synaptic long-term depression (LTD) with changes in synaptic development and plasticity, resulting in ASD and memory impairment [50,51,52].

Tamouza R and colleagues [53] analyzed the distribution of HLA haplotypes among Swedish children with ASD with and without regression and observed that the HLA-DPA1*01-DPB1*04 sub-haplotype was less represented in patients with regressive autism compared to those without regression, suggesting a protective role of this haplotype in regression.

Frye RE et al. [54] studied a group of subjects with ASD who were recruited for a natural history study. They found that being exposed to higher doses of PM_2.5_, either because of higher pollution levels or longer exposure to relatively lower PM_2.5_ doses as a consequence of outdoor recreational activities, could be correlated to an increase in regression. The authors speculated that this could have been due to the negative effect of pollution on an already (genetically) fragile mitochondrial physiology in children with ASD.

Although the mechanism leading to regression in children with ASD is still unclear, its negative prognostic role has been clearly demonstrated. Recently, Martin-Borreguero et al. [55] showed that children with RO-ASD had significantly higher core autistic symptoms and lower linguistic and social skills at 24-month follow-up. These children were also characterized by significantly worse adaptive functioning scores, both at diagnosis and 24 months later, and needed more intensive support and rehabilitative interventions. A large multicenter study that included 1027 children from China [56] confirmed these findings. Thompson et al. [57] were in line with these observations but also found no significant differences between children who had regressed from a normal functioning level and those who already had a reduced neurodevelopmental level.

Interestingly, a study conducted by Damiani et al. [58] in adults with ASD living in the rural area of Northern Italy found that those with RO-ASD showed worse behavioral profiles and more frequently received psychopharmacological prescriptions to compensate for clinical problems. Although the sample size of this study was too small to reach statistical significance, the trend was clear.

Finally, a skewed neuro-immune crosstalk may be involved early in the pathogenesis of different forms of autism, although only a few biomarkers have been identified. Increased B-cell expression of CD23+ along with different serum cytokine and adhesion molecule profiles in children with regressive ASD have been found [59]; further, a specific increase in IL-8 is described in infants with early autism compared to typical controls [60].

In conclusion, all the above research underlined clinical differences that potentially lead to a different need of rehabilitative and supportive interventions and point to a different etiology for regressive and non-regressive ASD. Our results are in line with them, suggesting a contribution for *SNAP-25* genetic variants in determining the early or the regressive onset of ASD and, in the last case, probably via an epigenetic pathway.

The most relevant limitation of this study is the relatively small cohort of patients with ASD for each group coupled with the lower percentage of patients with RO-ASD compared to those with EO-ASD. Future studies should include larger samples and, possibly, have a prospective design in order to prevent retrospective recall biases when trying to define a child as having RO-ASD or EO-ASD [9]. Also, the early serum concentration of the *SNAP-25* protein is needed to confirm different expression levels between EO- and RO-ASD as well as the evaluation of the miRNAs that target the *SNAP-25* gene.

## 4. Material and Methods

### 4.1. Patients and Controls

One hundred and twenty-seven children (93 males, 34 females, mean age 7.9 ± 4.3 years) with ASD diagnosis according to the DSM-5 criteria [1] were enrolled at the IRCCS Mondino Foundation National Neurological Institute of Pavia (Italy) and at the Complex Operating Unit of Child Neuropsychiatry, University of Sassari (Italy). Sixty-five children (51 males, 14 females, mean age 5.8 ± 2.9 years) were from Italian peninsular descent; the remaining 62 children were of Sardinian ancestry (42 males, 20 females, mean age 10.1 ± 4.5 years). Among them, 69 children (51 males and 18 females, mean age 8 ± 4.7 years) were clinically classified as early-onset ASD, and 58 children (41 males and 17 females, mean age 7.8 ± 3.9 years) were clinically classified as regressive-onset ASD based on the loss of previously acquired social–communicative skills. The mean age in months of regression onset was 41.3 ± 22.5.

Clinical, psychiatric, neurological, and neuropsychological evaluations were carried out in all children with ASD. The diagnosis was confirmed through clinical evaluation and the Autism Diagnostic Observation Schedule 2 (ADOS-2) [61]. Other diagnostic tools included the structured parent’s interview Autism Diagnostic Interview-Revised (ADI-R) [62] and the Childhood Autism Rating Scale (CARS) [63]. The global cognitive status was evaluated with Leiter Intelligence Scales [64], Wechsler Intelligence Scales [65], and Raven’s Progressive Matrixes [66]. The Children’s Global Assessment Scale (CGAS), which provides a measure of the impact on global functioning for youths under the age of 18, was used to evaluate the children’s general functioning [67].

The inclusion criteria were (a) age between 3 and 12 years and (b) a diagnosis of ASD confirmed by both clinical, not standardized, evaluation and the Autism Diagnostic Observation Schedule 2 [67]. The exclusion criteria were the diagnosis of psychotic disorders and/or intellectual disability and/or other developmental disabilities without ASD, according to the DSM-5 criteria. Patients with an ascertained lesion of the central nervous system and/or genetic syndrome were also ruled out from the study. Genotyping of the children with ASD was blinded to the clinical assessment.

Two hundred and five unrelated, healthy Italian subjects were enrolled as a control group. They were sex-matched with the patients with ASD (165 males and 40 females). Since only genetic data were used to compare cases and controls, no age matching was used.

The study was designed and conducted according to the Declaration of Helsinki; the research protocol was approved by the Don Gnocchi Foundation Ethical Committee (protocol n. 06_18/05/2016) on 18 May 2016.

### 4.2. SNPs Genotyping

Genomic DNA from patients and controls were isolated from peripheral blood or saliva. DNA from blood was obtained using a standard phenol/chloroform procedure, whereas from saliva, the ORAgene-DNA (DNA Genotek, Ottawa, ON, Canada) was used. The *SNAP-25* SNPs were estimated with real-time allelic discrimination using TaqMan Assay probes (Applied Biosystems, Carlsbad, CA, USA). For rs363039, rs363043, rs363050, and rs1051312, respectively, the C_327976_10, C_2488346_10, C_329097_10, and C_339356_20 Human Pre-Designed Assays (Applied Biosystems) were used. A total of 10 ng of DNA/sample was amplified in a 10 µL reaction using TaqMan Genotyping Master Mix on 96-well plates with a CFX96^TM^ System (Bio-Rad, Hercules, CA, USA). In each experiment, control samples of known genotypes and one negative control were included.

### 4.3. Statistical Analysis

The Hardy–Weinberg equilibrium (HWE) for the rs363039, rs363043, rs363050, and rs1051312 SNPs was calculated using a chi-square method in both cases and control. 2x2 contingency tables were used to compare the distribution of the rs363039, rs363043, rs363050, and rs1051312 alleles between the EO-ASD group, RO-ASD group, and healthy controls. Allele, genotype, and haplotype distributions between the groups were compared using 2X N contingency tables. A 2X2 contingency table was applied when a statistically significant result was found, and the resulting *p*-value was corrected for the degree of freedom (DF). The association of each polymorphism/genotype with different onsets was evaluated with the odds ratio (OR) and its 95% confidence interval (CI). *p*-values were considered significant when <0.05 and, if applied, after Bonferroni correction for the proper degrees of freedom (*Pc*). The associations of the EO-ASD and RO-ASD groups and of *SNAP-25* polymorphisms/haplotypes with the scores of clinical, behavioral, and functioning scales were tested with the parametric ANOVA (analysis of variance) or non-parametric Kruskal–Wallis and Mann–Whitney tests, depending on the fitness of the scales to the normal distribution measured with the non-parametric Kolmogorov–Smirnov test.

Data were analyzed with SPSS version 28.0 (IBM Corp. in Armonk, NY) and the open source openEpi https://www.openepi.com, accessed on 10 April 2023. Haplotype and linkage disequilibrium analyses were performed with SHEsis online version, http://analysis.bio-x.cn/myAnalysis.php, accessed on 10 April 2023, [68].

## 5. Conclusions

The etiopathogenesis of regression in ASD has not yet been clearly defined. Apart from those children with evidence of a mitochondrial dysfunction or with the onset of an epileptic encephalopathy, it is widely believed that many genetic and epigenetic factors are likely to be involved [23]. There are, however, very few studies that compare the genetic variants in children with regressive and non-regressive ASD. By studying *SNAP-25* gene variations, we described a different distribution for two different SNPs (rs363039 and rs1051312) between the two groups. Our data may indicate a different etiopathogenesis for the distinct ASD onsets, being that the regressive ASD is probably triggered by epigenetic factors that could affect the embryo–fetal as well as the post-natal development, as described in a recent model of autism risk [69]. *SNAP-25* is essential for pre- and post-synaptic transmission along with spine morphogenesis; with all these fundamental roles in the normal neurodevelopment, variation in its genotype can be associated not only with ADHD, bipolar disorder, and schizophrenia but also with different patterns of ASD onset.

These two *SNAP-25* variants may be useful as predictive genetic and diagnostic markers both for an early rehabilitation of rs1051312 carriers and for a careful monitoring of their neurodevelopmental trajectory, thus enabling a personalized intervention.

## Figures and Tables

**Figure 1 ijms-24-14042-f001:**
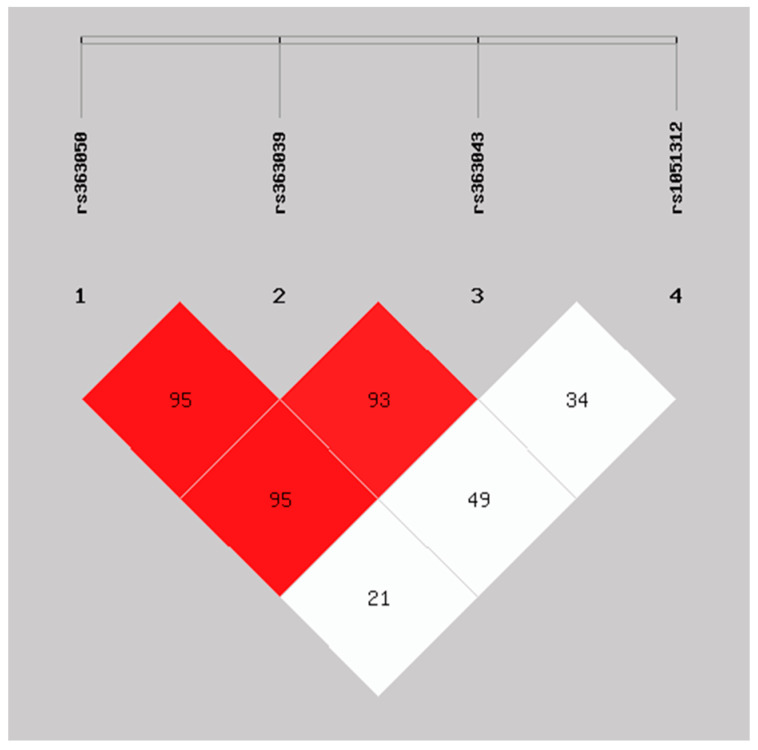
Linkage disequilibrium (LD) plot for the *SNAP-25* SNPs rs363050, rs363039, rs363043, and rs1051312. Linkage disequilibrium (LD) plot generated with Shesis software for the four analyzed *SNAP-25* SNPs *rs363050*, *rs363039*, *rs363043* (all in intron 1), and *rs1051312* (3′UTR). LD is displayed as pairwise D’ values, (*D*’). LD is represented as a gradient of colors with a red-to-white gradient reflecting higher-to-lower D’ values.

**Table 1 ijms-24-14042-t001:** Allele and genotype distributions of *SNAP-25* rs363050, rs363039, rs363043, and rs1051213 SNPs in children with ASD and healthy controls (HC).

Allele Frequency	Continental ASD N (%)	Sardinian ASD N (%)	*p* *	Total ASD N (%)	HC N (%)	*p* °
Rs363050						
A	79 (61.0)	75 (61.0)		154 (61.0)	244 (60.0)	
G	51 (39.0)	49 (39.0)	0.9	100 (39.0)	166 (40.0)	0.8
Rs363039						
G	97 (75.0)	81 (65.0)		178 (70.0)	257 (63.0)	
A	33 (25.0)	43 (35.0)	0.1	76 (30.0)	153 (37.0)	*0.05*
Rs363043						
C	94 (72.0)	84 (68.0)		178 (70.0)	298 (73.0)	
T	36 (28.0)	40 (32.0)	0.4	76 (30.0)	112 (27.0)	0.4
Rs1051312						
T	97 (75.0)	103 (83.0)		200 (79.0)	301 (73.0)	
C	33 (25.0)	21 (17.0)	0.1	54 (21.0)	109 (27.0)	0.1
Total	130	124		254	410	
**Genotype frequency**						
Rs363050						
AA	24 (37.0)	26 (42.0)		50 (39.0)	70 (34.0)	
AG	31(48.0)	23(37.0)		54(43.0)	104 (43.0)	
GG	10 (15.0)	13 (21.0)	0.4	23 (18.0)	31 (19.0)	0.3
Rs363039						
GG	35 (54.0)	28 (45.0)		63 (50.0)	78 (38.0)	
GA	27 (42.0)	25 (40.0)		52 (41.0)	101 (49.0)	
AA	3 (5.0)	9 (14.0)	0.1	12 (9.0)	26 (13.0)	0.1
Rs363043						
CC	35 (54.0)	32 (54.0)		67 (53.0)	111 (54.0)	
CT	24 (37.0)	20 (32.0)		44 (35.0)	76 (37.0)	
TT	6 (9.0)	10 (16.0)	0.5	16 (13.0)	18 (9.0)	0.5
Rs1051312						
TT	36 (55.0)	42 (68.0)		78 (61.0)	108 (53.0)	
TC	25 (38.0)	19 (31.0)		44 (35.0)	85 (41.0)	
CC	4 (6.0)	1 (2.0)	0.2	5 (4.0)	12 (6.0)	0.2
Total	127	127		127	205	

N: absolute number of alleles/genotypes, %: allele/genotype frequency, *: Comparison between continental and Sardinian ASD, °: Comparison between total ASD and healthy controls (HC). For 2xN contingency tables, *p* values were adjusted with the Bonferroni correction.

**Table 2 ijms-24-14042-t002:** Allele and genotype distributions of *SNAP-25* rs363050, rs363039, rs363043, and rs1051312 SNPs in children with ASD with early and regressive onsets.

Allele Frequency	ASD Early Onset N (%)	ASD Regressive Onset N (%)	*p*	HC N (%)	*p*
Rs363050					
A	86 (62.0) *	68 (59.0)°		244 (60.0)	
G	52 (38.0)	48 (41.0)	0.5	166 (40.0)	* 0.5. ° 0.8
Rs363039					
G	103 (75.0) *	75 °(65.0)		257 (63.0)	
A	35 (25.0)	41 (35.0)	0.08	153 (37.0)	* 0.009, ° 0.7
Rs363043					
C	91 (66.0) *	87 (75.0)°		298 (73.0)	
T	47 (34.0)	29 (25.0)	0.12	112 (27.0)	* 0.1. ° 0.6
Rs1051312					
T	99 (72.0) *	101 (87.0)°		301 (73.0)	
C	39 (28.0)	15 (13.0)	0.003	109 (27.0)	* 0.7, ° 0.002
Total	148	116			
**Genotype frequency**					
Rs363050					
AA	30 (43.0) *	20(34.0)°		70 (34.0)	
AG	26(38.0)	28(48.0)		104 (43.0)	
GG	13 (19.0)	10 (17.0)	0.47	31 (19.0)	* 0.2. ° 0.5
Rs363039					
GG	40 (58.0) *	23° (40.0)		78 (38.0)	
GA	23 (33.0)	29 (50.0)		101 (49.0)	
AA	6 (9.0)	6 (10.0)	0.11	26 (13.0)	* 0.015, ° 0.9
Rs363043					
CC	32 (46.0) *	35 (60.0)°		111 (54.0)	
CT	27 (39.0)	17 (29.0)		76 (37.0)	
TT	10 (14.0)	6 (10.0)	0.29	18 (9.0)	* 0.3. ° 0.5
Rs1051312					
TT	34 * (49.0)	44 (76.0)°		108 (53.0)	
TC	31 (45.0)	13 (22.0)		85 (41.0)	
CC	4 (6.0)	1 (2.0)	0.008	12 (6.0)	* 0.9, ° 0.006
Total	69	58		205	

N: absolute number of alleles/genotypes, %: allele/genotype frequency, *: Comparison between the early onset group and HC, °: Comparison between the regressive onset group and HC. For 2xN contingency tables, *p* values were adjusted with the Bonferroni correction.

**Table 3 ijms-24-14042-t003:** *SNAP-25* rs363039 and rs1051312 under the genetic Dominant model associations with the ASD early and regressive onsets and healthy controls (HC).

Genotype Frequency	ASD Early Onset N (%)	ASD Regressive Onset N (%)	*p*	OR (95%IC)	HC N (%)	*p*	OR (95%IC)
Rs363039							
GG	40 (58.0) *	23 (40.0)			78 (38.0)		
GA/AA	29 (42.0)	35 (60.0)	0.04	2.09 (1.02–4.3)	127 (62.0)	* 0.004	2.2 (1.28–3.9)
rs1051312							
TT	34 (49.0)	44 (76.0)°			108 (53.0)		
TC/CC	35 (51.0)	14 (24.0)	0.002	3.2 (1.5–6.9)	97 (47.0)	° 0.001	2.8 (1.47–5.6)
N	69	58			205		

N: absolute number of genotypes, %: genotype frequency, *: Comparison between the early onset group and HC, °: Comparison between the regressive onset group and HC.

**Table 4 ijms-24-14042-t004:** *SNAP-25* rs363050, rs363039, rs363043, and rs1051312 haplotype analysis in the early onset ASD (EO-ASD), regressive onset ASD (RO-ASD), and healthy controls (HC).

	ASD Early Onset		ASD Regressive Onset				HC		
	N	%	N	%	*p*	OR (95%IC)	N	%	*p*
**4 SNPs HAPLOTYPE**									
A-G-T-C	24 *	17.5	5	4.2	0.0008	4.7 (1.8–14.1)	30	7.2	* 0.0006
G-A-C-T	28	20.3	36°	31.0	0.05	1.7 (1.9–3.1)	101	24.7	° 0.17
**2 SNPs HAPLOTYPE**									
G-C	33 *	23.8	12	10.5	0.004	2.7 (1.3–5.7)	72	17.5	* 0.1
A-T	29	20.9	39°	33.3	0.02	1.9 (1.1–3.3)	116	28.3	° 0.3
Total N	138		116				410		

N: number of haplotypes, *: Early onset group compared to HC, °: Regressive onset group compared to HC. For 2xN contingency tables, *p* values were adjusted with the Bonferroni correction.

## Data Availability

The data presented in this study are available upon request from the corresponding author.

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
