# Peer review of "The Role of SNAP-25 in Autism Spectrum Disorders Onset Patterns"

_ijms, 2023, doi:10.3390/ijms241814042_

Round 1
Reviewer 1 Report
Dear authors,
The manuscript entitled "The Role of SNAP-25 in Autism Spectrum Disorders Onset Patterns" by Bolognesi is one of the interesting articles about the role of SNAP-25 in ASD. This manuscript is well written with new information on early and regressive ASD of SNAP-25 and its polymorphism. To improve the manuscript quality, the authors should address the comments below.
- The title matches the findings. However, proper Capitalisation should be used in the title.
- The introduction part is well described, with adequate information and relevant literature on SNAP-25 protein. Lines 62-65 represent the more general role of SNAP-25. Mention that the SNAP-25 polymorphism is associated with cognition and ASD.
- The SNARE complex genes name should be mentioned correctly in lines 66-68.
- The previous findings of the SNAP-25 polymorphism should be briefly described to understand why this study is essential in human.
- The tables convey the major results of this study. In table, due to non-significant results in Allele frequency and Genotype frequency of these polymorphism in the selected cohort, the authors do not mention the description of the table. The notes should be mentioned for the table 1. Also, mention the p value correction (i.e Bonferroni calculation) wherever applicable, must be presented in italics (line 132, etc).
- SNAP-25 was in italics some places (line 196) and some places not in italics (line 194). Correct them all.
- The authors used a small cohort for this present study. Hence, limitations should be added.
Author Response
The manuscript entitled "The Role of SNAP-25 in Autism Spectrum Disorders Onset Patterns" by Bolognesi is one of the interesting articles about the role of SNAP-25 in ASD. This manuscript is well written with new information on early and regressive ASD of SNAP-25 and its polymorphism. To improve the manuscript quality, the authors should address the comments below.
- The title matches the findings. However, proper Capitalisation should be used in the title.
A: The title was changed as per your suggestion in: "The Role of SNAP-25 in Autism Spectrum Disorders Onset Patterns"
- The introduction part is well described, with adequate information and relevant literature on SNAP-25 protein. Lines 62-65 represent the more general role of SNAP-25. Mention that the SNAP-25 polymorphism is associated with cognition and ASD.
A: To answer point 2 we added a brief description of previous findings of SNAP-25 associations with ASD:
“The SNAP-25 rs363043 and rs363050 polymorphisms were shown to be associated with hyperactivity and cognitive deficits in Italian ASD children, respectively [27, 28], whereas in the Han Chinese population, the rs363018 and rs8636 predicted a higher likelihood to develop ASD [29]. Finally, the rs3746544 G allele was observed to be more frequently carried by Iranian ASD children [30].”
- The SNARE complex genes name should be mentioned correctly in lines 66-68.
A: We amended the description of the SNARE complex genes as follows:
“SNAP-25, VAMP (vesicle-associated membrane protein), and Syntaxins constitute the SNARE complex: these proteins are essential for the exocytosis of synaptic vesicles and thereby for synaptic transmission”
- The previous findings of the SNAP-25 polymorphism should be briefly described to understand why this study is essential in human .
A: We added a description of SNAP-25 polymorphisms in other disorders:
“In addition, the SNAP-25 gene has been largely associated with different neurological and no neurological disorders. SNAP-25 rs363050 (A) and rs363043 (T) alleles, as well as the rs363050/rs363043 A-T haplotype, were associated with Alzheimer's Disease [31] Similarly the rs363050 AA genotype was significantly associated to the risk of Sarcopenia, [32] whereas SNAP-25 rs1051312, was suggested to play a role in the development of Multiple Sclerosis [33].
- The tables convey the major results of this study. In table, due to non-significant results in Allele frequency and Genotype frequency of these polymorphism in the selected cohort, the authors do not mention the description of the table. The notes should be mentioned for the table 1. Also, mention the p value correction (i.e Bonferroni calculation) wherever applicable, must be presented in italics (line 132, etc).
A: Thank you very much for these observations:
We added the missing notes in Table 1:
“N: absolute number of alleles /genotypes, %: allele /genotype frequency, *: Comparison between Continental and Sardinian ASD, °: Comparison between Total ASD and Healthy Controls (HC).
We also added to all table notes, when applicable, the sentence:
“For 2xN contingency tables, p values were adjusted with the Bonferroni correction.”
And, as per your suggestion, we corrected the “p” value in Italics “p” in the whole paper.
- SNAP-25 was in italics some places (line 196) and some places not in italics (line 194). Correct them all.
A: We checked all the “SNAP-25” and corrected them in italics.
- The authors used a small cohort for this present study. Hence, limitations should be added.
A: We better clarified the limitations of our study for the small cohort as follows:
“ The most relevant limitation of this study is the relatively small cohort of ASD patients for each group coupled with the lower percentage of the RO-ASD with respect to the EO-ASD counterpart. Future studies should include larger samples and, possibly, have a prospective design, in order to prevent retrospective recall biases when trying to define a child as a RO-ASD or an EO-ASD [9]. Also, the early serum concentration of the SNAP-25 protein is needed to confirm different expression levels between EO- and RO-ASD as well as the evaluation of the miRNAs that target the SNAP-25 gene “

Reviewer 2 Report
Dear authors,
A very interesting article is presented to the community. A relevant topic has been discussed about the role of SNAP-25 in the appearance of ASD. In general, the article is well articulated, but here are some suggestions for improvement.
1. The title adequately reflects the object of study and the topic to be investigated.
2. The abstract is presented in an organized manner. The background, the method, the results and some brief conclusions are correctly presented.
3. Keywords are concise.
4. In the introduction, it is recommended to update the DSM-5 citation, given that in 2022 said manual was updated:
- American Psychiatry Association [APA] (2022). Diagnostic and Statistical Manual of Mental Disorders (DSM-V-TR), 5th Edition. American Psychiatry Association.
5. The introduction presents sufficient previous studies and the information is written in a clear and orderly manner. It is recommended to include previous more current studies. In the introduction, of 36 studies cited, only 11 are current (2018-2023).
6. A "Method" section must be included before the "Results", not at the end. In general, the "Methods" section is well done.
7. The title of the tables must be centered, as indicated by the MDPI regulations.
8. The letter "p" of lines 90, 100, 103, etc. It must be presented in italics. Review the rest of the document to correct possible errors.
9. The reference to table 2 must be included in the wording, not isolated on line 111. This also happens in table 3 (line 121), table 4 (line 137) and figure 1 (line 145).
10. Table 1 must have a note to properly understand the meaning of the initials, as has been done in the rest of the tables. Review when these initials should be in italics and when not.
11. The results are presented in an ordered manner.
12. The discussion relates the findings of this research to previous studies, but these do not belong to the current panorama. It is strongly recommended to update the studies cited by other more current ones, which belong to the last five years. If this were not possible, it would be interesting to include it in the limitations of the study.
13. It would be enriching to include a prospective research as a solution to the sample limitation indicated by the authors.
14. It is recommended to expand the "Conclusions" section.
15. The study was carried out in accordance with the guidelines of the Declaration of Helsinki and was approved by the institutional ethics committee.
16. The references must be adapted to the MDPI regulations.

Author Response
A very interesting article is presented to the community. A relevant topic has been discussed about the role of SNAP-25 in the appearance of ASD. In general, the article is well articulated, but here are some suggestions for improvement.
1.The title adequately reflects the object of study and the topic to be investigated.
- The abstract is presented in an organized manner. The background, the method, the results and some brief conclusions are correctly presented.
- Keywords are concise.
- In the introduction, it is recommended to update the DSM-5 citation, given that in 2022 said manual was updated:
- American Psychiatry Association [APA] (2022). Diagnostic and Statistical Manual of Mental Disorders (DSM-V-TR), 5th Edition. American Psychiatry Association.
A: Thank you very much for your recommendation, we corrected the reference for the DSM-5 citation.
- The introduction presents sufficient previous studies and the information is written in a clear and orderly manner. It is recommended to include previous more current studies. In the introduction, of 36 studies cited, only 11 are current (2018-2023).
A : Thank you very much for your suggestion about the inclusion of more current studies. When it was possible, we updated some articles, now the current studies (2018-2023) are 19.
- A "Method" section must be included before the "Results", not at the end. In general, the "Methods" section is well done.
A: As per your suggestion we moved the “Material and method” section before the “Results” one.
- The title of the tables must be centered, as indicated by the MDPI regulations.
A: Thank you for the remark, now the Titles of the tables are centered.
- The letter "p" of lines 90, 100, 103, etc. It must be presented in italics. Review the rest of the document to correct possible errors.
A: We checked the whole paper and all the “p”s were corrected in italics.
- The reference to table 2 must be included in the wording, not isolated on line 111. This also happens in table 3 (line 121), table 4 (line 137) and figure 1 (line 145).
A: We included the references to Tables in the text as you suggested.
- Table 1 must have a note to properly understand the meaning of the initials, as has been done in the rest of the tables. Review when these initials should be in italics and when not.
A: We are so sorry for the omission, now, we added the missing information about the meaning of the initials for Table 1, as follows:
“N: absolute number of alleles /genotypes, %: allele /genotype frequency, *: Comparison between Continental and Sardinian ASD, °: Comparison between Total ASD and Healthy Controls (HC). “
- The results are presented in an ordered manner.
- The discussion relates the findings of this research to previous studies, but these do not belong to the current panorama. It is strongly recommended to update the studies cited by other more current ones, which belong to the last five years. If this were not possible, it would be interesting to include it in the limitations of the study.
A: Thank you for your comments, we added these more recent studies (2018-2023) specifically regarding clinical studies and two about the possible neuroimmune involvement in ASD :
Frye RE et al [62] studied a group of ASD subjects recruited for a natural history study. They found that being exposed to higher doses of PM2.5, either because of higher pollution levels or of longer exposure to relatively lower PM2.5 doses as a consequence of outdoor recreational activities, could be correlated to an increase in regression. The authors speculated that this could have been due to the negative effect of pollution on an already (genetically) fragile mitochondrial physiology in children with ASD.
Although the mechanism leading to regression in ASD children is still unclear, its negative prognostic role has been clearly demonstrated. Recently, Martin-Borreguero et al [63] showed that children with RO-ASD had significantly higher core autistic symptoms and lower linguistic and social skills at 24-month follow-up. These children were also characterized by significantly worse adaptive functioning scores, both at diagnosis and 24 months later, and needed more intensive support and rehabilitative interventions. A large multicenter study including 1027 children from China [64] confirmed these findings. Thompson et al [65] were in line with these observations but also found no significant differences between those children who had regressed from a normal functioning level and those who already had a reduced neurodevelopmental level.
Interestingly, a study conducted by Damiani et al [66] in ASD adults living in the rural area of Northern Italy found that those with RO-ASD showed worse behavioural profiles and more frequently received psychopharmacological prescriptions to compensate for clinical problems. Although sample size of this study was too small to reach statistical significance, the trend was clear
Finally, a skewed neuro-immune cross-talk may be involved early in the pathogenesis of different forms of autism, although only a few biomarkers have been identified (54): increased expression of CD23+ B-cell along with different serum cytokine and adhesion molecules profiles of regressive ASD children have been found [67]; further, a specific increase of IL-8 is described in infants with early autism compared typical controls [68].
In conclusion, all the above researches underlined clinical differences potentially leading to a different need of rehabilitative and supportive interventions, and point to a different etiology for regressive and non regressive ASD. Our results are in line with them, suggesting a contribution for SNAP-25 genetic variants in determining the early or the regressive onset of ASD, and in the last case, probably, via an epigenetic pathway.
- It would be enriching to include a prospective research as a solution to the sample limitation indicated by the authors.
A: Thank you very much, we followed your recommendation to encompass the problem regarding the small cohort:
“The most relevant limitation of this study is the relatively small cohort of ASD patients for each group coupled with the lower percentage of the RO-ASD with respect to the EO-ASD counterpart. Future studies should include larger samples and, possibly, have a prospective design, in order to prevent retrospective recall biases when trying to define a child as an RO-ASD or an EO-ASD [9]. Also, the early serum concentration of the SNAP-25 protein is needed to confirm different expression levels between EO- and RO-ASD as well as the evaluation of the miRNAs that target the SNAP-25 gene.”
- It is recommended to expand the "Conclusions" section.
A: In line with your suggestion we expanded the “Conclusion” section as follows:
- CONCLUSIONS
“ The etiopathogenesis of regression in ASD has not yet been clearly defined. Apart from those children with evidence of a mitochondrial dysfunction or with the onset of an epileptic encephalopathy, it is widely believed that many genetic and epigenetic factors are likely to be involved [23]. There are however very few studies that compare genetic variants in children with regressive and nonregressive ASD. By studying SNAP-25 gene variations, we described a different distribution for two different SNPs (rs363039 and rs1051312) between the 2 groups. Our data may indicate a different etio-pathogenesis for the distinct ASD onsets, being the regressive ASD probably triggered by epigenetic factors, which could affect the embryo-fetal as well as the post-natal development, as described in recent model of autism risk [69]. SNAP-25 is essential for pre- and post-synaptic transmission along with spine morphogenesis; with all these fundamental roles in the normal neurodevelopment, variation of its genotype can be associated not only with ADHD, Bipolar disorder and schizophrenia but also with different patterns of ASD onset.
These two SNAP-25 variants may be useful as predictive genetic and diagnostic markers both for an early rehabilitation of rs1051312 carriers and for a careful monitoring of their neurodevelopmental trajectory, thus enabling a personalized intervention.”
- The study was carried out in accordance with the guidelines of the Declaration of Helsinki and was approved by the institutional ethics committee.
- The references must be adapted to the MDPI regulations.
A: We are so sorry for the mistake. Now we corrected the references according to the MDPI reference guide.

Round 2
Reviewer 2 Report
Dear authors,
All suggestions for improvement have been rigorously addressed. The article has been significantly improved.
Thank you.